# Increasing the Working Time of Forging Tools Used in the Industrial Process of Producing a Disk-Type Forging Assigned for a Gearbox through the Application of Hybrid Layers

**DOI:** 10.3390/ma17123005

**Published:** 2024-06-19

**Authors:** Marek Hawryluk, Łukasz Dudkiewicz, Jacek Borowski, Jan Marzec, Roger Tkocz

**Affiliations:** 1Department of Metal Forming, Wroclaw University of Science and Technology, Lukasiewicza 5, 50-371 Wroclaw, Poland; lukasz.dudkiewicz@pwr.edu.pl (Ł.D.); jan.marzec@pwr.edu.pl (J.M.); 2Łukasiewicz Research Network—Poznań Institute of Technology, 61-139 Poznan, Poland; jacek.borowski@pit.lukasiewicz.gov.pl; 3MAHLE Behr Ostrów Wielkopolski, 63-400 Ostrów, Poland; roger.tkocz@mahle.com

**Keywords:** die forging, tool durability, hybrid layers, disk type forging

## Abstract

The article discusses the phenomena and destructive mechanisms occurring on the surface of 1.2344 steel dies used during the hot forging of disc-type forgings. Preliminary research has shown that gas nitriding alone, used so far, is insufficient due to the occurrence of destructive mechanisms other than abrasive wear, such as thermal and thermomechanical fatigue, which cause the average durability of such tools to be approximately 5000 forgings. Analyses were also carried out to assess the load on forging tools using numerical modeling (Forge 3.0NxT), which confirmed the occurrence of large and cyclically changing thermal and mechanical loads during the forging process. Therefore, in order to increase operational durability, it was decided to use two types of hybrid layers, differing in the PVD coating used: TiCrAlN and CrN, and then subjected to gas nitriding (GN). The obtained results showed that, depending on the area of the tool and the current working conditions, the applied PVD coatings protect the surface layer of the tool against the dominant destructive mechanisms. In both cases, the strength increased to the level of 7000 forgings, the tools could continue to work, and globally, slightly better results were obtained for the GN+TiCrAlN layer. The CrN-type layer protects the tool more against thermal fatigue, while the TiCrAlN layer is more resistant to abrasive wear. In areas where the hybrid layer was worn, a decrease in hardness was observed from 1300 HV to 600–700 HV, and in places of intense material flow (front—point 2 and tool bridge—point 9) the hardness dropped to below 400 HV, which may indicate local tempering of the material. Moreover, the research has shown that each process and tool should be analyzed individually, and the areas in the tool where particular destructive mechanisms dominate should be identified, so as to further protect the forging tool by using appropriate protective coatings in these areas.

## 1. Introduction

The high intensity of wear in tools used in die-forging processes puts more and more focus on problems connected with die durability. Forging tools, despite the significant technological progress and the development of science, are still characterized by an unstable and relatively low durability, which significantly affects the costs of production as well as the quality of the forgings [1]. For this reason, forging tools are exposed to the operation of many destructive factors, which causes their wear, both prematurely and after a longer operation time. The durability of tools used in forging processes is affected by very many factors: the conditions under which the forging process is run (the temperature and geometry of the tools and the preform, the processing speed, the type and amount of lubricant ensuring optimal tribological conditions, etc.). The durability of the dies also depends on the material used for the tools, their proper design and construction, the appropriate thermal treatment, as well as the applied surface layer modification [2]. The process of tool wear is very complex and the tool destruction is caused by many mechanisms operating together and often being opposing in character. These processes include not only the friction phenomena and the related abrasive and adhesive wear, fretting and oxidation, but also other processes, such as brittle cracking, adhesion, etc. The most common main destructive mechanisms include abrasive wear [3,4] and plastic deformation in warm forging [5,6,7] as well as plastic deformation in hot forging [8,9], thermo-mechanical cracking [10,11,12] and thermo-mechanical fatigue [13,14]. Among them, the most thoroughly analyzed is the wear mechanism, which occurs mostly in cold forging processes [15,16], but also in hot forging processes; although, despite common opinion, it is not a dominating destructive mechanism [17]. The literature provides many studies referring to a complex analysis of the main tool destruction mechanisms in hot die-forging processes [18,19,20]. The listed forging tool destruction mechanisms mainly refer to the tools’ surface layer. And so, modifying the properties of the surface layer of the forging tools is the most effective way to improve their durability. In order to avoid these phenomena, different methods are applied, which consist of introducing new solutions into the process itself, ensuring control and stability, as well as producing and depositing proper protective layers or applying alternative materials, the purpose of which is the protection from one or several destructive mechanisms [21,22,23,24].

One of the methods of improving the durability of forging tools is gas nitriding (GN) and plasma nitriding (PN), which increase the abrasion resistance, fatigue strength and corrosion resistance. Gas nitriding is a low-temperature (usually 520 °C) thermo-chemical treatment process causing small deformations performed to improve the surface tribological properties of finished or almost-finished steel elements. In the case of adding a gas containing carbon, the process is called gas carbonitriding. The source of the nitrogen and carbon atoms “in statu nascendi” are ammonia and carbon dioxide. The created nitrided layer contains two basic zones: the nitride zone ε (an interstitial compound with a hexagonal lattice), and ε + γ’, γ’ (an interstitial compound with a regular lattice). As it does not reveal its granular structure during etching, this zone is commonly called the “white layer”. Right under the nitride zone, there is another diffusive zone (an interstitial solid nitrogen solution in iron—α) with the possibility of the occurrence of nitride precipitates. The thin layer of nitride compounds formed on the element’s surface is responsible for improving the wear and abrasion resistance (tribological properties) as well as increasing the corrosion resistance. The diffusive layer serves to improve the fatigue strength and works as a substrate for the hard layer of compounds, shaping the optimal characteristics of the stress distribution on the cross-section of the whole layer. The modern installations of gas nitriding make it possible to precisely control and shape the structure of the obtained nitrided layers. The control and regulation of the process with the use of technically advanced atmosphere analyzers, mass flow controllers and uniform atmosphere circulation ensures achieving an optimal layer for the real operation conditions of the element [25,26]. Unfortunately, the application of a nitrided layer does not always give a clear effect of durability improvement as it does not efficiently protect the tool’s surface layer from the destructive factors occurring during operation, such as thermal shocks, intensive friction and high mechanical loads. Another method of increasing forging tool durability is the use of physical vapor deposition (PVD). PVD coatings limit the abrasive wear and reduce the intensity of damage caused as a result of thermo-mechanical fatigue. A flaw of these coatings, however, is their cracking and loss of cohesion with the substrate. This negative phenomenon is caused by the plastic deformation of the surface layer occurring right after a short time of the tool’s operation. Then, loose hard particles of the coating are formed, which, working as an abradant, significantly accelerate the abrasion wear. Due to the increasing requirements referring to the tool quality and the application of more and more advanced die-forging processes, there is a search for new methods of increasing tool life. Tests conducted at many research centers in the scope of complex surface engineering methods have led to the creation of hybrid technologies, consisting of the application of two or more surface engineering techniques. The best effects have been observed for hybrid methods that combine thermal treatment methods and the PVD technique. The most commonly applied hybrid layers are layers of the PN+PVD type. The main task of a nitrided layer is to increase the hardness and plastic deformation resistance of the substrate. This protects the hard PVD coating from a loss of internal cohesion to the substrate. In turn, PVD coatings constitute an effective insulation of the substrate, thus limiting the effect of external destructive factors in the operation process. This said, in the case of the use of hybrid layers, the key is to obtain a proper intermediate layer between the nitrided layer and the coating as, in the case of a big difference in hardness between the two, we can observe their rapid separation and cracking. The literature contains works showing that the use of this type of hybrid layer (PVD) increases the operational durability of tools in hot forging processes, both on presses [27,28] and on dies used on hammers [29]. There are also works in which the authors used PACVD coatings of the TiB2 and TiCN type [30], as well as TiN/TiAlN and TiN type coatings used on dies intended for forging crankshafts [31]. You can also find articles using multi-layer coatings (Cr/CrN)x8 [32], Cr/CrN, Cr/CrN/AlCrTiN and CrN/AlCrN/AlCrTiSiN [33] and many others. In the case of a deeper analysis, at the level of microstructures, it is worth considering the possible impact of temperature on phase transitions and changes in morphology at the micro-level and whether this may affect the quality of the protective coating used [34,35]. Owing to a combination and mutual interaction of different technologies, we can obtain surface layer properties that are not achievable by means of other techniques applied separately [36]. Most research works, including those of the authors of this article, have shown that by applying coatings to a nitrogen-rich substrate, durability can be increased by an average of 40–80% and even more. It seems that the current application of hybrid layers to improve forging tool durability is fully justified by the effectiveness of these methods as well as the scientific and economical aspects. However, each process, and even a single tool, should, despite many similarities, be analyzed individually, because only the directions are known, but each process is slightly different, which means that specific coatings will work well in one process and give the opposite effect in another. Only by carrying out a series of tests for a specific tool, consisting of an analysis of the entire technology on a macro scale, as well as detailed tests on a micro-scale, supported by numerical simulations, will a full and in-depth analysis of consumption be possible. Therefore, such a dedicated, individual approach to operational durability is new and will allow finding the optimal solution for a given application in the form of a hybrid layer.

## 2. Materials and Methods

The test subject is an industrial hot forging process conducted to produce a disk-type forging, which, after mechanical treatment (to create the toothing), constitutes the front wheel of the reverse gear in motorcars, e.g., Opel, as the key element of the gearbox. The die-forging process itself is realized on a crank press with the pressure of 25 MN in three operations. The first one is upsetting, the second is preliminary die forging, and the third is finishing forging. The initial temperature of the charge material equals 1150–1180 °C (forging temperature). The material for the forging is steel QS1920S0. In the case of die-forging processes, an important issue is the hardness of the tools (operation time) described based on the number of forgings produced on a given tool. In the analyzed case, the tools are made of steel WCLV. Table 1 presents the chemical composition of both materials.

For the second and third operations, the tools used up to that point are subjected to nitriding to obtain a hardness of 1100–1200 HV. The thickness of the nitrided layer equals about 0.2 mm. The average tool durability in the particular operations is as follows: I operation—about 30,000 forgings, II operation—about 5000 forging and III operation—20,000 forgings. For this reason, due to their lowest durability, a detailed analysis was performed on the tools from the second operation, in order to improve their performance time. For the tools used in operation II, a hybrid layer of the nitride layer/PVD coating type was used. It was decided to apply two variants of the coating: CrN (chromium nitride) and TiCrAlN. Table 2 shows the parameters of the production of the hybrid layer as well as the properties of the nitrided layer and the CrN coating.

In order to evaluate the applied hybrid layers and analyze the industrial forging process, the following were performed: a complex analysis of the forging technology with the use of, e.g., a thermovision camera Flir 840 (FLIR Systems, Inc. Wilsonville, OR, USA) and a fast camera Casio Pro EX-F1 (Casio, Tokyo, Japan), as well as a macroscopic analysis of the tools by means of Cannon EOS 60D (Cannon, Ōita, Japan). A 3D scanning with the use of an optical scanner GOM ATOS II (GOM, Braunschweig, Germany) was also carried out. The numerical simulations were made by numerical packed Forge 3.0 NxT (Transvalor, Biot, France). The observations of the changes on the working surface were conducted by means of a scanning electron microscope (SEM) with magnifications of over 1000× with the use of TESCAN VEGA 3 (TESCAN GROUP, a.s., Brno–Kohoutovice, Czech Republic coupled with an EDX detector. In turn, the microstructural tests of the surface layer of the tool’s cross-section were performed by the light microscopy method after the tools had been etched in by means of a stereoscopic microscope (Keyence VHX-S600E, Osaka, Japan) and a light microscope (Olympus BX51M, Tokyo, Japan). The microhardness tests of the cross-section in the function of the distance from the surface were realized with the use of a microhardness tester LECO. The performance tests of the tools were conducted under industrial conditions in one of the selected die forges in Poland. In the first place, after producing 3000 forgings, all the dies were subjected to preliminary analysis (a macroscopic analysis and a wear measurement by means of an optical scanner). The dies were returned to the production process and another 4000 forgings were produced. Detailed investigations of the wear mechanisms for these dies were performed after a total of 7000 forgings had been produced. A macroscopic analysis as well as a microscopic analysis of the tool surface and hardness measurements were conducted in order to compare the effectiveness of both coatings.

## 3. Results and Discussion

### 3.1. Analyze of Industrial Process

The technology of forging in open dies applied in the analyzed process is realized in a manual system, which means that the charge material as well as the preforms and the forgings in the consecutive operations are relocated by the blacksmith operator. The time of one cycle equals 12–14 s. Forging in a manual system causes certain instability of the process as well as lack of repeatability; however, in the case of production problems connected with precise manipulation by the robots, the operation by a human is fully justified as the operator is able to quickly respond and make fast decisions on the spot. This is important especially when the work of the forging aggregate is interrupted or when a forging falls out of the tool, in which case the situation can be controlled quickly by the operator, contrary to an automatized system and robotized work. Figure 1 shows a view of the forging process, a preform and a forging obtained in the particular operations, and also worn lower die inserts. The thermovision measurements were performed by means of a camera in order to determine the temperature distributions on the tool surfaces in the particular operations of the analyzed process (Figure 1d).

The highest temperatures are present on the die in the first operation. On the die in the second operation, we can see a clear difference in temperature for the whole surface. The section located “on the side” of the blacksmith is about 100 °C cooler than the section on the side that is the furthest from the “blacksmith”. This is most probably caused by an improper cooling technique (adjustment of the nozzles) or a non-uniform placing of the forging in the die by the blacksmith. The maximal die temperature for the second operation equals about 270 °C. On the die in the third operation, we can observe a more uniform course of temperatures than for the die from the second operation. The maximal temperature there is about 250 °C. The tools are heated to the desired temperature by means of a heated waste material with a temperature of about 1100 °C, and the heating time is about 1–1.5 h. For the cooling and lubrication of the tools, a lubricant in the form of a 1:20 graphite–water mixture is used, which is fed in an automated way by a specially designed cooling–lubricating device. The level of the lubricant is checked every 100 items or every 2 h. It is forbidden to apply a continuous blast on the die inserts.

The point of reference for the introduction of hybrid layers on the tools was an analysis of the durability of the tools used so far in the second operation, which were only subjected to nitriding. Figure 2 presents exemplary results for the tools after different numbers of produced forgings together with one of the most frequent defects, in the form of the so-called “tool material wash-out”, which is a detachment of larger parts/fractions as a result of extreme operation conditions as well as the presence of destructive mechanisms. As we can notice in the case of the tools used so far—the lower die inserts only subjected to nitriding (Figure 2)—clear wear can be observed only for dies that have produced 3000 forgings. The wear is localized both in the central part of the die next to the pusher and in the vicinity of the bridge, where the forging material enters the groove for the flash. However, we should point out that the wear in the central part is non-uniform and there is no “smooth transition” into the neighboring areas.

A probable cause of such a non-uniform wear was the nitrided layer, which, in this area, is exposed to intensive overheating, which causes its local tempering, and as a result of high cyclic pressures, we observe detachment of larger parts of the tool material. In the other areas, the observed wear, especially ^o^ in the vicinity of the bridge, is uniform. This said, this is an area of an intensive flow of the deformed material and it is dominated by abrasive wear, which is counteracted by the nitrided layer. Additionally, for a more thorough analysis, microhardness measurements were made of the analyzed die on the section of 0.3 mm from the working surface into the material (Figure 3).

The analysis was performed for the bridge area, that is at the place where the deformed forging material intensively flows out of the working impression into the flash (the area marked with a circle in Figure 2c). The microhardness measurements showed that the nitrided layer after producing 5000 forgings is still stable (Figure 3a). This proves the effectiveness of nitriding as a way of protection from the disadvantageous abrasive wear. In turn, a small drop of hardness to about 800–900 HV can be explained by the operation of temperature. For this reason, the use of an additional protective layer in the form of a coating in this case is fully justified. The presented results point to a full “wear off” of the nitrided layer. Only for a small number of forgings (Figure 2a—about 1000 forgings) can we observe and conclude about a partial presence of the nitrided layer in these areas, which was also confirmed by the microstructural tests.

### 3.2. Numerical Simulations

In order to determine the most vulnerable areas of the forging tool to damage as a result of destructive mechanisms, numerical simulations were performed using a computational package (Forge 3.0NxT) in the axisymmetric state of deformation for a thermomechanical model with deformable tools (die inserts—elements with heat exchange). Such software is dedicated to analyzing and simulating hot die-forging processes [37]. The geometry of the tools, the blank, and other technological parameters of the process were introduced into the program based on CAD models and technological cards. A crank press with a pressure of 25 MN was adopted. The speed of punch movement depended on the angular position of the press. The temperatures of the tools were measured with a thermal imaging camera, thanks to which their temperatures during the forging process were determined, respectively: 1st operation—550 °C, 2nd operation—250 °C, 3rd operation—270 °C. The times of subsequent operations were determined using a camera capable of recording 400 frames per second. The recorded average forging cycle of one forging (3 operations) was 12.5 s. The following values of friction factors were assumed: between the dies and the deformed material 0.3; between punches and deformable material 0.3 and between ejectors and deformable material 0.2. Material data, i.e., thermal expansion, specific heat, thermal conductivity, dependence of the course of Young’s modulus on temperature, Poisson’s ratio for the tool material: 1.2364 (WCLV) steel were adopted from the MATILDA material database. The plastic properties of the forged material were determined in an upsetting test on the GLEEBLE simulator. Based on the results of the initial FEM simulation, it was determined that the strain rate of the batch varies from 0.1 to 100 s^−1^ and the temperature ranges from 600 to 1200 °C, the determined curves were entered into the program. The heat transfer coefficients in contact with the surroundings were assumed to be 25 and 0.35 kW/m^2^K, respectively. Figure 4a shows the forging forces as a function of the rotation angle obtained from numerical modeling in individual operations, while Figure 4b presents a comparison of shape and material flow from FEM and real forging.

In the force diagram for the three operations, it can be seen that the force in the first operation is small (semi-free upsetting) compared to the forces in the second operation (pre-die forging) and the third operation (finish forging—calibration). The highest force occurs in the second forging operation and amounts to over 26 MN, which exceeds the permissible force of an industrial press. Such a high value of force may result from mistuning of the FEM model (e.g., friction forces); however, talks with the technology and design department of the forging company confirm that the second forging operation is critical and may result in machine overload. The analyzed fiber distribution on the cross-section of the forging and its shapes in subsequent operations allow us to conclude that the developed numerical model of the process is correct and reliably presents the material flow in the industrial forging process (Figure 4b). The force values also determine, in a way, the average tool life in individual operations. Figure 5a shows the temperature field distributions in the tools used in the second operation in the final phase of deformation. In turn, Figure 5b shows the distributions of equivalent stresses.

The maximum temperature of over 650 °C (Figure 5a) occurs on the central surface of the lower die insert (the contact point of the blank from the first operation) and on the radii of the blank. Based on the presented distributions in the inserts (Figure 5b), it can be observed that in places in the corner (characteristic holes marked in green), where the geometry of the cut causes the concentration of equivalent stresses, the values reach as much as over 1100 MPa. Please remember that the results refer to a single forging operation. It can be assumed that if the simulations were carried out for cyclic loading (e.g., several thousand repetitions taking into account changes after each operation), then microcracks would appear in the indicated places, which would propagate and lead to the complete destruction of the tool over time. The large temperature gradients and amplitudes observed in the numerical modeling (from 250 °C to 600 °C) and the high pressures and forces occurring during the forging process, as well as the intense material flow, may cause the destruction of tools (geometry changes) due to local tempering of the material and the appearance of plastic deformation. In turn, in places of stress concentration resulting from the shape of the tool, cracks may additionally appear as a result of mechanical fatigue.

Additionally, in order to find the cause of the premature removal of the nitrided layer in the inserts from the second operation, a decision was made to analyze and inspect its production more closely (Figure 6).

Figure 6 shows photographs of the microstructures for the lower die insert from the second operation after it has produced 3000 forgings and the microstructure of the pilot from the nitriding process, as well as the arrangement of the tool in the retort. For the areas where the forging material was intensively deformed and stayed in contact with the tool surface for a long time, we can observe damage to the nitrided layer (Figure 6b). Also of concern is the small size of the layer on the pilot (Figure 6c) and the method of arrangement of all the tools in the retort (Figure 6d). The arrangement can also hinder the diffusion of nitrogen into the surface layer of the particular tools, which, in effect, can lead to non-uniform nitriding or even lack of a nitrided layer in some areas. The performed analysis of the technology, with a special consideration of the dies used in II forging operation, showed that the forging tools work under difficult operation conditions (high cyclic alternating mechanical and thermal loads). Moreover, the observations and analyses showed that the forging tools are degraded mainly by a few dominating destructive mechanisms, which include abrasive wear, plastic deformation, thermal and thermo-mechanical fatigue, as well as mechanical cracking. This said, the presence and intensity of the particular mechanisms depend on the area of the tool, and also, the occurrence of these mechanisms changes with the course of their operation. The identified difficult working conditions of the tools show that the use of only the durability-increasing method applied so far, in the form of nitriding, is insufficient, and in order to increase durability, it is advisable to undertake further steps and methods, e.g., the application of hybrid layers of the nitrided layer-coating type.

### 3.3. Macroscopic Analysis

For a macroscopic evaluation of the quality state of the tools and determination of the loss in the particular areas of the analyzed tools, an optical scanner GOM ATOS II was used. The data collected from the scanned working surfaces of the dies after their operation were compared with the original shape of the tools before their work. The results in the form of colored maps with deviations with respect to the nominal dimension, i.e., the state of the dies before the work, have been presented in Figure 7 and Figure 8. The wear of the tool, after it has produced 3000 forgings, takes place mainly in the central part and on the radius of the exit onto the bridge for the flash. In the central part, the highest wear reaches the value of m and is located next to the opening for the pusher. In this area, the tool, due to the manner of lubrication, is the most exposed to the mechanism of thermal fatigue, and so, the coating should be characterized by a low heat exchange coefficient. The wear in this area of the CrN coating (Figure 7), compared to TiCrAlN (Figure 8), is much higher after the tool has produced both 3000 and 7000 forgings.

The die with the TiCrAlN coating, after 3000 forgings have been produced, demonstrates higher wear in the area of the bridge for the flash. However, together with the increase in the number of produced forgings (7000), the wear in this area decreases compared to the tool with the CrN coating. The initial better abrasive wear resistance on the radius of the bridge (CrN), due to the formation of a network of thermo-mechanical cracks on its surface, worsened together with the increasing number of produced forgings. The CrN coating cracks and its loose particles work as an abradant, drastically increasing the wear of the tool. A worse crack resistance of the CrN coating increases the abrasive wear of the whole die.

### 3.4. Microscopic Tests

The analysis of the changes in the tool surface layer was performer by means of a scanning electron microscope. The die impression was divided into five areas. Figure 9 shows two views from each tool area for both coating variants. The first area is characteristic of a non-uniform wear. It is dominated by two destructive mechanisms. In the central part, where the preform is placed, we observe thermal fatigue of the subsurface layer. For both dies, we can see a network of fatigue cracks, which, however, are different in character.

In the case of the CrN coating, in a magnified image, we can see the primary network and a very fine secondary network covering the whole surface as well as the edges of the primary cracks. The spalled hard particles of the coating move between the charge material and the die. A result of this is the formation of grooves on the surface, both in a section of area 1 and the beginning of area 2, where the material flows intensively. The working surface of the dies in the third area looks similar (Figure 10).

Here, no wear occurs. In area 5, on the radius, we observe the mechanical fatigue of the tool. For both coatings, the development of the crack is similar. Also in the vicinity of the gap, similar spalling of the coating takes place. In turn, in the area of the bridge for the flash and the radius preceding the bridge, in the case of the CrN coating, the appearing grooves are much deeper than in the case of the TiCrAlN coating. This obviously results from the previously described cracking character of the CrN coating and the negative effect on the wear.

Additionally, in order to confirm the presence of a hybrid layer of the GN+coatingtype, microstructural tests were performed. Figure 11 shows the area in the tool with the CrN coating, which includes the radius of the exit onto the die’s bridge. The wear in this area is uniform on the whole tool circumference. We observe a truncation of the radius at this place and a decrease in the bridge’s width (Figure 11a). At the place of the radius, grooves are formed in the radial direction, coming from the abrasive wear. These grooves are much smaller than in area 1. On the cross-section, we can also see traces of a crack network formed with a smaller number of forgings, poorly visible on the surface due to being sealed over by oxides and rub-offs (Figure 11b). Beyond the radius in the bridge area, we can see a spalling of the network (Figure 11c), which, as shown by the chemical composition microanalysis, is formed by the PVD coating (Figure 12).

Figure 12 shows the results of the X-ray microanalysis EDS (Energy Dispersive Spectrometry), which point to the presence of a CrN-type PVD coating, whereas the hardness measurements also confirmed the presence of a nitrided layer under the CrN coating.

### 3.5. Hardness Measurements

The microhardness measurements by the Vickers method (with the loading force of 0.98 N) were made on the cross-section in the direction from the working surface into the tool material. The hardness distribution for the lower die from the second operation after it produced 7000 forgings is shown in Figure 13.

In the presented graphs (Figure 12), it can be observed that for the variant with a TiCrAlN coating at points 1, 2, 3, 4, the coating showed better insulating properties. These areas are mainly exposed to thermal fatigue and material tempering. The coating at these points allowed for a smaller decrease in hardness than in the case of the CrN coating. TiCrAlN shows greater resistance of the coating to abrasive wear at points 9 and 10. In areas 5, 6, 7, 8, the values are similar, which is due to the small share of destructive mechanisms. Smaller decreases in hardness and greater depths of the nitrided layer indicate better operational properties of the TiCrAlN multilayer coating.

## 4. Summary

The article presents the research results referring to the possibilities of increasing the durability of die inserts used in the forging process of producing a disk-type forging (assigned for the gearbox) through the application of hybrid layers of the nitrided layer+PVD coating type. In the technology applied so far, the used tools are made of steel 1.2364, which are only subjected to nitriding, and as demonstrated by the tests, the application of only this method of durability increase (thermo-chemical treatment in the form of nitriding) is insufficient. This has been confirmed by the observations and the complex analysis of technology, which showed that the nitrided layer works effectively in the case when the dominating destructive mechanism is abrasive wear. The average durability of such tools equaled about 5000 forgings. In turn, as shown by the tests, in the analyzed process, other destructive mechanisms also occur, which significantly lower the tool’s durability, especially in the areas where there is no intensive flow of the deformed material, resulting in abrasive wear. Additionally, for a more complete analysis, numerical modeling was carried out to assess the load condition of forging tools in particular. FEM results confirmed high mechanical and thermal loads. The comprehensive analysis of the industrial forging process allowed for the decision to use a method of increasing durability consisting of the use of hybrid layers for tools, such as GN+TiCrAlN and GN+CrN. The tools were introduced into the production process in order to evaluate the effect of the applied layers on their wear.

## 5. Conclusions

The tests carried out in industrial conditions and research allowed for drawing the following detailed conclusions:The most dangerous factor for the analyzed tools with the applied hybrid layers is abrasive wear in the vicinity of the bridge, whereas in the central area of the tool, thermal fatigue is dominant.It was also established that another main dominating destructive mechanism is thermo-mechanical fatigue, which, beyond the bridge area, occurs in almost every other area of the tool with different intensities, and with the highest intensity occurring on the central part. Such a mechanism leads very quickly to the formation of microcracks. The further development of these cracks is conditioned by the process parameters, the contact time and the material’s flow rate, and it usually leads to the formation of a crack network on the whole contact surface.It was observed that this network can be stable during the further forging process and the wear occurs as the material is washed away from the crack area, and a secondary crack network can also be formed, which undergoes spalling. Considering this, these phenomena require explaining in the physical sense because, according to the mechanics of spalling, an old crack should more easily expand than a new crack should form.In the case of prolonged contact of the hot forging with the tool surface in its central area, we can observe plastic deformation as a result of local tempering of the material in this area.Based on the macroscopic analyses, including 3D scanning and microscopic observations, it was demonstrated that the key factor responsible for the hybrid layer increasing the wear resistance of the tool is the characteristics of the cracking of the applied coating.The high and variable temperature gradients and amplitudes observed in FEM (from 250 °C to 600 °C) as well as high pressures in the tools and forces occurring during the forging process indicate a highly probable destruction of the tools.The tests showed a bigger effect of brittle cracking on the wear of the die in the case of the CrN coating. In turn, the hybrid layer with the TiCrAlN coating demonstrates better performance parameters.Both applied coatings protect the tool from the operation of high temperatures much better than a nitrided layer alone, that is, they constitute a thermal barrier, yet they exhibit a low resistance to high pressures.The conducted research has shown that each process and tool should be analyzed individually, and the areas in the tool dominated by particular destructive mechanisms should be identified, so as to further protect the forging tool by using appropriate protective coatings in these areas.

## Figures and Tables

**Figure 1 materials-17-03005-f001:**
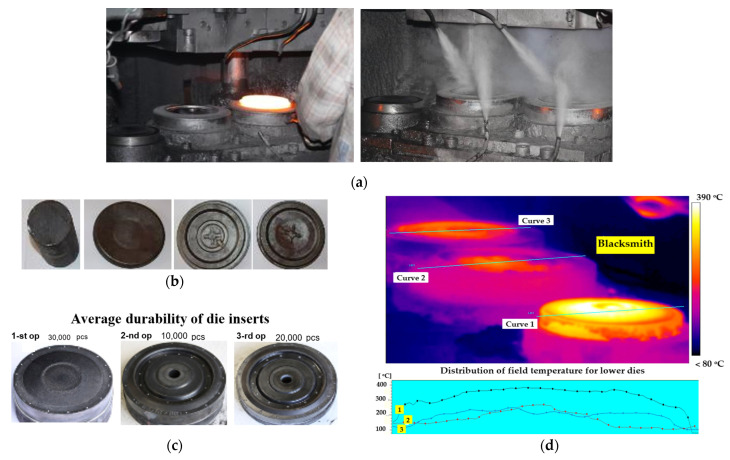
The view of (**a**) the forging process of disk-type forging during forming (**left**) and lubrication (**right**) by water with graphite at 1:20 concentration, (**b**) the initial billet and the forgings in individual operations, (**c**) the bottom die inserts from operations: I 1st—upsetting, 2nd—initial forging, 3rd—finishing forging, (**d**) the thermogram from the temperature field distribution.

**Figure 2 materials-17-03005-f002:**
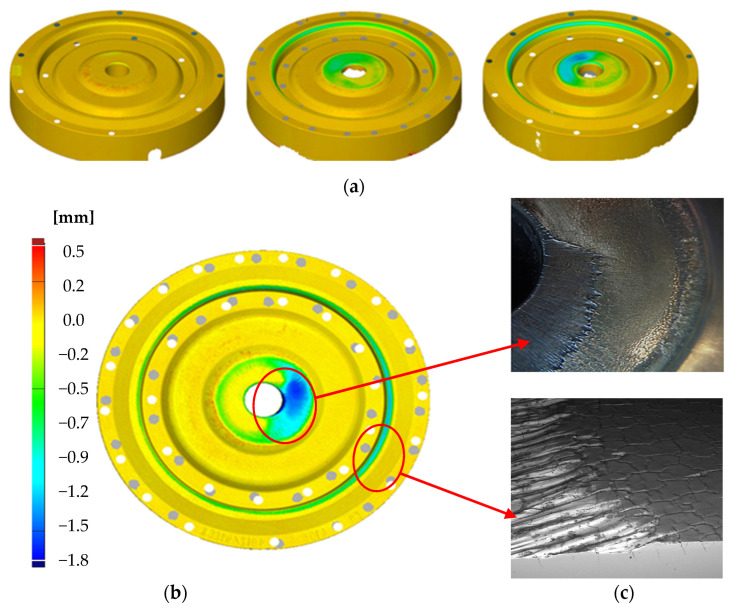
Analysis of exemplary tools from the second operation after different numbers of produced forgings: (**a**) scans of the dies after 1000, 3000 and 5000 forgings, (**b**) a front view of a worn tool removed from production after producing 5000 forgings, (**c**) magnified macroscopic views of the very worn areas in the central part and near to the bridge of die.

**Figure 3 materials-17-03005-f003:**
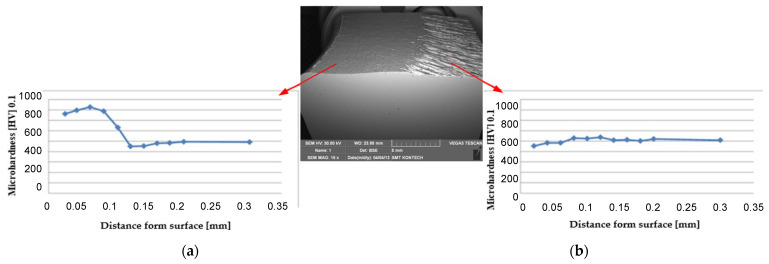
Microhardness profiles HV 0.1 from the area: (**a**) with a nitrided layer and (**b**) without a hybrid layer.

**Figure 4 materials-17-03005-f004:**
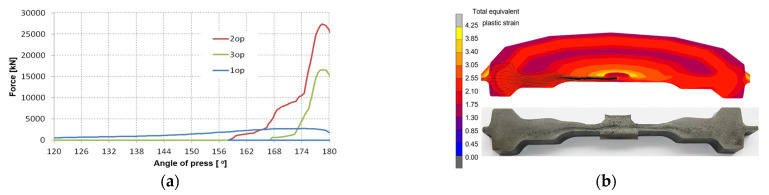
The FEM results: (**a**) forging forces obtained as a function of the crankshaft rotation angle of the press, (**b**) distribution of plastic strains after 2 operations (**up**) and fiber distribution in the forging—Jacewicz test (**down**).

**Figure 5 materials-17-03005-f005:**
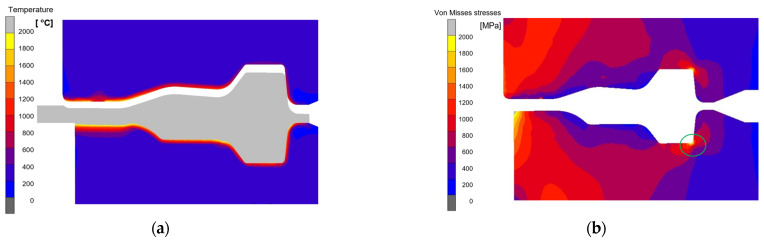
The FEM results with distribution of (**a**) temperature in Celsius, (**b**) equivalent stresses in MPa, the circle marked in green indicates the possibility of stress concentration and cracking occurring in this area due to mechanical fatigue.

**Figure 6 materials-17-03005-f006:**
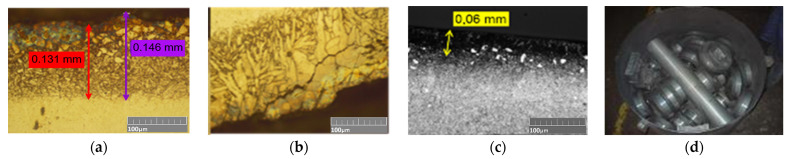
Photographs of: (**a**) the microstructure of a die insert after producing 3000 forgings for the non-working part together with a measurement of the nitrided layer thickness, (**b**) visible damage to the nitrided layer, (**c**) the “pilot” from nitriding, (**d**) the arrangement of all tools in the retort.

**Figure 7 materials-17-03005-f007:**
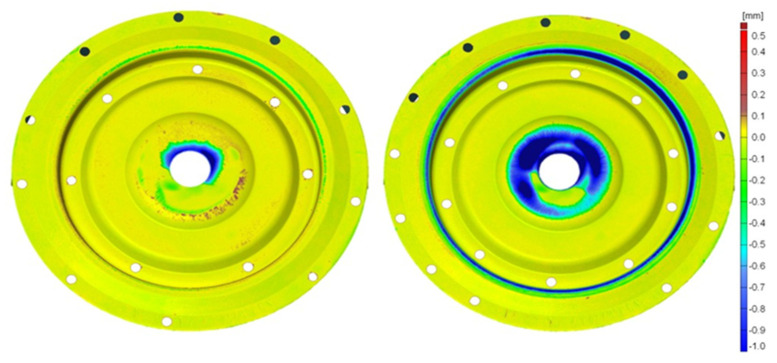
Wear of a die with a hybrid layer (CrN coating) obtained from the optical scanner, from the left: after producing 3000 and 7000 forgings—second operation, lower die.

**Figure 8 materials-17-03005-f008:**
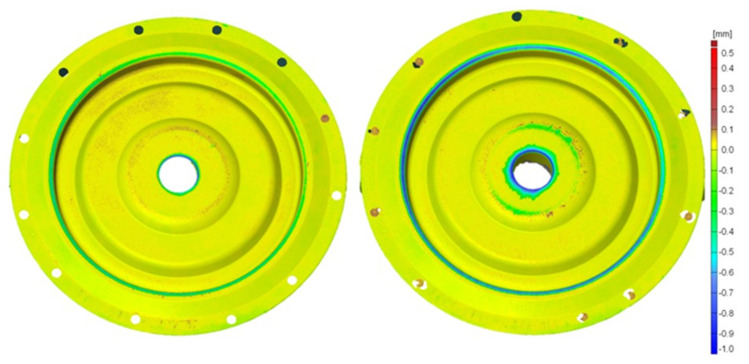
Wear of a die with a hybrid layer (TiCrAlN coating) obtained from an optical scanner, from the left: after producing 3000 and 7000 forgings—second operation, lower die.

**Figure 9 materials-17-03005-f009:**
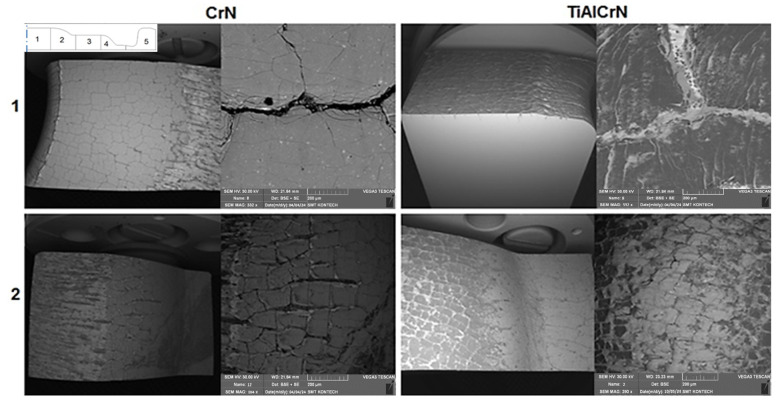
A view of the particular areas (1–2) of the lower die from the second operation subjected to a microscopic observation, in the upper left corner there is a cross-sectional diagram of the tool with the indicated five areas where the analysis was performed.

**Figure 10 materials-17-03005-f010:**
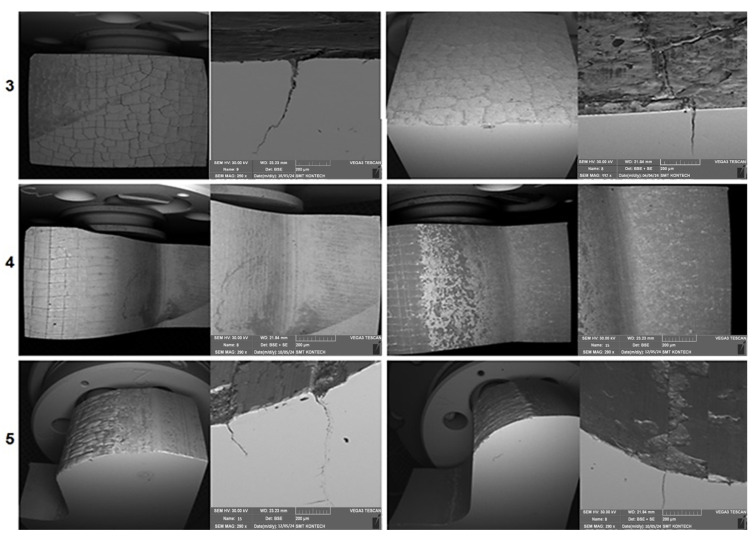
A view of the particular areas (3–5) on the lower die from the second operation subjected to a microscopic observation.

**Figure 11 materials-17-03005-f011:**
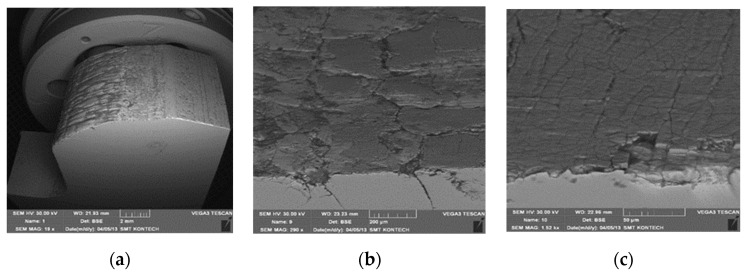
The working surface of the analyzed die in area 3: (**a**) visible decrease in the bridge’s width, (**b**) visible oxides and rub-offs, (**c**) spalling of the network.

**Figure 12 materials-17-03005-f012:**
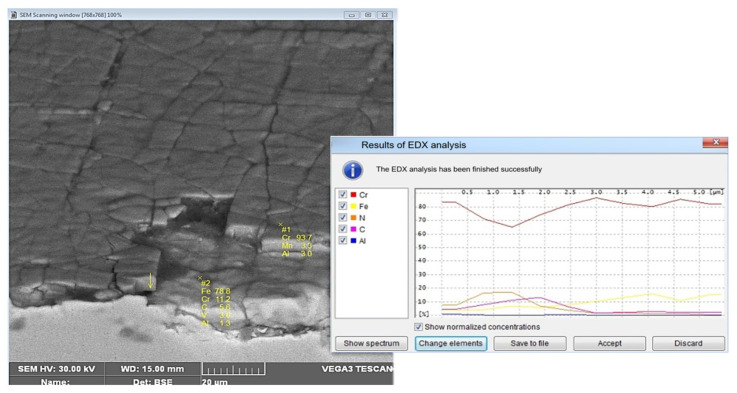
A chemical composition analysis in the bridge area of the die.

**Figure 13 materials-17-03005-f013:**
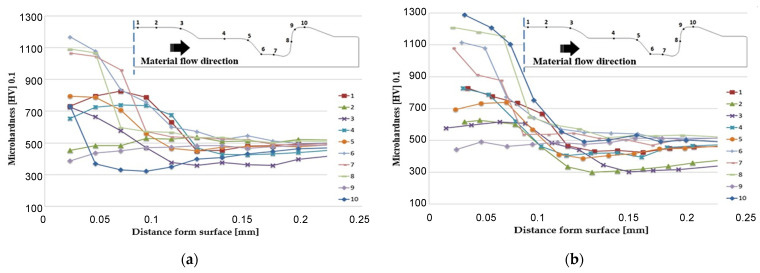
The hardness distribution on the lower die from the second operation after producing 7000 forgings: (**a**) the CrN coating and (**b**) the TiCrAlN coating.

**Table 1 materials-17-03005-t001:** Chemical composition of forging and tool steel.

Steel/Components	C	Mn	Si	P	S	Cr	Ni	Mo	V	W	Cu
QS1920 (1.4131)	0.15–0.18	1.25	0.18	Max. 0.016	Max. 0.03	Max. 1.03	Max. 0.12	Max. 0.03	0.004	-	Max. 0.21
WCLV (1.2344)	0.35–0.45	0.2–0.5	0.8–1.2	Max. 0.03	Max. 0.03	4.5–5.5	Max. 0.35	1.2–1.5	0.8–1.1	Max. 0.30	Max. 0.30

**Table 2 materials-17-03005-t002:** The parameters of: a nitrided layer and a hybrid layer (nitrided layer/CrN and TiCrAlN coating).

Parameters of the Nitrided Layer		
Phase structure	Diffusive zone	
Surface hardness	1000 HV0.5	
Effective thickness of an 800 HV hardness zone	g_800_ = 0.13 mm	
Nitriding parameters	T = 520 °C Atmosphere: 90% H_2_ + 10% N_2_ *p* = 4.3 mbar	
**Parameters of a hybrid layer**	**GN/CrN coating**	**GN/TiCrAlN coating**
Thickness	g ≈ 3.8 μm g ≈ 5.28 μm	g ≈ 6.7 μm
Hardness	H = 24 ± 1.6 GPa	H = 30 ± 2.1 GPa
Young modulus	E = 279 ± 16 GPa	E = 337 ± 12 GPa
Friction factor—steel	f = 0.32	f = 0.48
Roughness	Ra/Rz/Rt = 0.43/1.16/1.92	Ra/Rz/Rt = 0.29/2.28/3.40
Adhesion	FnC1 = 46 N, FnC2 = 60 N, FnC3 = 103 N	FnC1 = 15 N, FnC2 = 83 N, FnC3 = 101 N

## Data Availability

Data are contained within the article.

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
