# Peer review of "Increasing the Working Time of Forging Tools Used in the Industrial Process of Producing a Disk-Type Forging Assigned for a Gearbox through the Application of Hybrid Layers"

_materials, 2024, doi:10.3390/ma17123005_

Round 1
Reviewer 1 Report
Comments and Suggestions for Authors
In the manuscript entitled “Increasing the working time of forging tools used in the industrial process of producing a disk type forging assigned for a gear box through the application of hybrid layers,” the authors present several interesting results. Despite this information, the following points must be addressed before I recommend accepting the manuscript for publication in the Materials MDPI journal:
1. The abstract must be rewritten to show how the research was done, what was obtained, and which observations were significant and their implications.
2. In Section 2, the chemical characteristics of the employed materials must be provided, as well as the brand and model of the instruments and equipment used. This is in order to guarantee reproducibility for the reader.
3. In Figure 1, some images are missing labels such as (a), (b), etc. It is important that all the figures in the manuscript present the corresponding labels in order to avoid confusing the reader and to maintain proper control in the description of the figure.
4. The discussion in the manuscript must be enhanced. The obtained results should be compared with other works, and the implications, differences, or similarities must be commented on and explained, respectively.
5. Particularly, it would be of great interest to add a paragraph in the manuscript commenting on how temperature can promote a phase transition and changes or modification of morphology at micro-level and discussing whether this phenomenon can affect the explored samples and how it might do so. I recommend citing and discussing the following articles:
H. Rojas-Chávez, H. Cruz-Martínez, et al., The formation of ZnO structures using thermal oxidation: How a previous chemical etching favors either needle-like or cross-linked structures, Materials Science in Semiconductor Processing, Volume 108, 2020, 104888, https://doi.org/10.1016/j.mssp.2019.104888
Cortés-Valadez, P.J.; Baños-López, et al., Bryophyte-Bioinspired Nanoporous AAO/C/MgO Composite for Enhanced CO2 Capture: The Role of MgO. Nanomaterials 2024, 14, 658. https://doi.org/10.3390/nano14080658
6. I recommend adding a paragraph before the conclusion section where the contribution of the work and its novelty are highlighted in comparison to other similar works.
Author Response
Dear Reviewer,
Thank you for your review and valuable comments.
Detailed responses to all questions are provided in a separate file.
best regards

Reviewer 2 Report
Comments and Suggestions for Authors
The paper elucidates the possibilities of forging resistivity via additional hybrid nitride layers and looks scientifically sound.
I want to draw attention to the following issues:
i) Are there any quantitative assessments for the ‘increasing the work time’ in a number of cycles added on or expected lifetime enhancement?
ii) The protective layer is of micron thicknesses, wherein the wear propagates on the millimeter scale (fig. 2, 7-8). Were there microstructure results representative at all? How do the coatings remain within such wear thicknesses?
iii) Provide a conclusion that matches the aim and title of the paper.
Author Response

(The authors gave the same response as above.)

Reviewer 3 Report
Comments and Suggestions for Authors
This is a very nice work. The paper is well organized, all methods are described adequately and the conclusions are supported by the results. Therefore, I recommend that the paper will be considered for publication in Materials. I suggest the following minor corrections for improvement of the paper.
- Introduction. Overal, the introduction provides a detailed discussion of the existing literature. However, the link to the goal of the present paper with respect to existing research gap is missing. The authors may add a short discussion in the last paragraph of the introduction.
- Some sloppy typos can be found in the paper, I suggest that the authors check carefully and correct them.
Comments on the Quality of English LanguageEnglish is fine. Minor editing of existing typos.
Author Response

(The authors gave the same response as above.)

Reviewer 4 Report
Comments and Suggestions for Authors
The authors investigated the effect of the hybrid coating to enhance the durability of forging tools. The detail analysis results will be useful to readers. However, there are several points to address:
・In the introduction, there is a lack of description of what is academically new about the authors' research compared to previous references.
・The scale bars in the microscopic images (Fig. 3, 6, 9, 10, 11 and 12) and vertical axes in Fig. 4 and 5 are unclear.
・It is unclear where areas 1 to 5 in Fig. 9 correspond to in the image in Fig. 8.
・The explanations for Fig. 9 and 10 are insufficient. (What does the left side of the column represent? What do the two images inserted in each photographs represent?)
・In Line3 from the bottom of page5, wasn't it the case after 5000 forges that the clear wear was observed?
・Are the positions indicated by the two red arrows in Fig. 3 appropriate? (cross section?)
Author Response

(The authors gave the same response as above.)

Reviewer 5 Report
Comments and Suggestions for Authors
The authors discuss their efforts to increase the durability of forging dies used in a hot forging process. They describe the phenomena occurring at the die’s surface (for the second hot forging operation), where a nitrided layer-PVD coating type was applied. Gas nitriding alone proved insufficient, due to the occurrence of thermal and thermo-mechanical fatigue, limiting average durability to the order of 5000 forgings. The authors performed an analysis to assess the load status of forging tools using numerical modeling. This analysis confirmed the occurrence of high, fluctuating thermal and mechanical loads during the forging process. To increase durability, they used two types of hybrid layers, differing in the applied PVD coating: TiCrAlN and CrN for dies made of steel 1.2367 and later subjected to gas nitriding. The following validation tests were carried out: a process analysis, 3D scanning of the tools, microstructural tests and hardness measurements. Depending on the tool area and operation conditions, the applied PVD coatings protect the surface layer of the tool from different dominant destructive mechanisms attaining an increase of durability to the level of 7000 forgings. Slightly better results were obtained for the GN+TiCrAlN layer.
The authors should address the following issues in an improved and carefully revised version of their manuscript:
Lines 56-57, Line 67 etc: Please avoid commenting of multiple reference citations at once.
Figure 1a: what is the cooling medium?
Figure 1d: please increase temperature graph size and resolution
Line 193: correct to “automated way”
Line 234. Insert citation for the software and short description of its modeling capacities.
Lines 249-250: please insert reference citation for database and simulator tool
Figure 5: increase font size in scale
Figures 7 – 8: Have you examined possible improvements of die design in the vulnerable area?
Line 300: Correct as: “the arrangement of all tools in the retort”
Figure 12: please increase size and resolution of the results of EDX analysis
Figure 13: increase font size and graph resolution.
Please insert a separate, concise Conclusions section as the last part of the paper.
Comments on the Quality of English LanguageThe manuscript should be carefully language - edited by a specialist.
Author Response

(The authors gave the same response as above.)

Round 2
Reviewer 1 Report
Comments and Suggestions for Authors
The authors have satisfactorily addressed the suggestions. Therefore, I recommend accepting the manuscript for publication.
Reviewer 5 Report
Comments and Suggestions for Authors
The revised version is acceptable for publication.
Comments on the Quality of English LanguageSome further English improvements required